# Peripheral intravenous catheter insertion and use of ultrasound in patients with difficult intravenous access: Australian patient and practitioner perspectives to inform future implementation strategies

Jessica A. Schults[1,2,3,4]*, Pauline Calleja[5], Eugene Slaughter[2], Rebecca Paterson[1,3,4], Claire M. Rickard[1,2,3,6,7], Catriona Booker[1,6], Nicole Marsh[1,3,6], Mary Fenn[3,6], Jenny Kelly[5,7], Peter J. Snelling[4,8,9,10], Joshua Byrnes[11], Gerben Keijzers[8,10,12], Marie Cooke[3]

1 The University of Queensland, School of Nursing, Midwifery and Social Work, St Lucia, Brisbane, Queensland, Australia, 2 Herston Infectious Diseases Institute (HEiDI), Metro North Hospital and Health Service, Herston, Brisbane, Queensland, Australia, 3 School of Nursing and Midwifery, Griffith University, Nathan, Brisbane, Queensland, Australia, 4 Child Health Research Centre, University of Queensland, Brisbane, Queensland, Australia, 5 Central Queensland University, Cairns, Queensland, Australia, 6 Nursing and Midwifery Research Centre, Workforce Development and Education Centre, Royal Brisbane and Women's Hospital, Herston, Brisbane, Queensland, Australia, 7 Health and Wellbeing Service Group, Townsville Hospital and Health Service, Townsville, Queensland, Australia, 8 Department of Emergency Medicine, Gold Coast University Hospital, Gold Coast, Queensland, Australia, 9 Sonography Innovation and Research (Sonar) Group, Gold Coast, Queensland, Australia, 10 School of Medicine and Dentistry, Griffith University, Southport, Queensland, Australia, 11 Centre for Applied Health Economics, Griffith University, Nathan, Brisbane, Australia, 12 Faculty of Health Sciences and Medicine, Bond University, Gold Coast, Queensland, Australia

* j.schults@uq.edu.au

## Abstract

### Objective

To understand healthcare worker and patient experience with peripheral intravenous catheter (PIVC) insertion in patients with difficult intravenous access (DIVA) including the use of ultrasound (US).

### Methods

Descriptive study using 1-on-1 semi-structured interviews conducted between August 2020 and January 2021. Purposeful sampling was used to recruit healthcare practitioners (HCPs) and patients with DIVA who had PIVC experience. Data were analysed using inductive thematic analysis. Interview data were than mapped to the implementation theory Behaviour Change Wheel to inform implementation strategies.

### Results

In total 78 interviews (13 patients; 65 HCPs) were completed with respondents from metropolitan (60%), regional (25%) and rural/remote (15%) settings across Australia. Thematic

restrictions with the sharing of the data set (Griffith University Human Research Ethics Committee) as data contains potentially sensitive information. Requests for access to de-identified data may be made to research-ethics@griffith.edu.au or the corresponding author j.schults@uq.edu.au and progress through the ethics committee.

**Funding:** This work was supported by the National Health and Medical Research Council (NHMRC) Partnership Project Grant (APP1180193). https://www.nhmrc.gov.au/ The funders had no role in study design, data collection and analysis, decision to publish, or preparation of the manuscript.

**Competing interests:** I have read the journal's policy and the authors of this manuscript have the following competing interests: Jessica Schults reports grants from Becton Dickinson unrelated to the current project. Claire Rickard: discloses that her current or previous employer has received on her behalf: investigator-initiated research grants from BD-Bard, Cardinal Health and Eloquest; and consultancy payments for lectures or opinion from 3M and BD-Bard; unrelated to current project. Nicole Marsh: reports that her affiliated universities have received on her behalf, speaker fees from 3M, investigator-initiated research grants from Becton Dickinson, Cardinal Health, Eloquest Healthcare and a consultancy payment from Becton Dickinson for clinical feedback related to peripheral intravenous catheter placement and maintenance (unrelated to the current project). Marie Cooke: discloses that her previous employer has received on her behalf: investigator-initiated research grants from BD-Bard unrelated to current project. PC, ES, RP, CB, MF, JK, GK, PS, JB, GK have no conflicts of interest to disclose.

analysis revealed 4 major themes: i) Harmful patient experiences persist, with patient insights not leveraged to effect change; ii) 'Escalation' is just a word on the front lines; iii) Heightened risk of insertion failure without resources and training; and iv) Paving the way forward–'measures need to be in place to prevent failed insertion attempts. Themes were mapped to the behaviour change wheel and implementation strategies developed, these included: staff education, e-health record for DIVA identification, DIVA standard of care and DIVA guidelines to support escalation and ultrasound use.

## Conclusion(s)

DIVA patients continue to have poor healthcare experiences with PIVC insertion. There is poor standardisation of DIVA assessment, escalation, US use and clinician education across hospitals. Quality, safety, and education improvement opportunities exist to improve the patient with DIVA experience and prevent traumatic insertions. We identified a number of implementation strategies to support future ultrasound and DIVA pathway implementation.

## Introduction

Approximately 90% of hospitalised patients receive a peripheral intravenous catheter (PIVC) [1], yet insertion is challenging, with two thirds of first attempt insertions failing and some patients requiring more than 10 insertion attempts (needlesticks) [2–5] to obtain access. Nationally, Difficult Intravenous Access (DIVA) affects 30–50% of hospitalised patients [6,7]. Patients at highest risk of DIVA typically fall within the age extremes [8,9], have chronic disease (resultant poor vein quality) [8,10], invisible and/or non-palpable due to excess adipose tissue [2,11]; or live in rural/remote areas, with limited access to advanced practitioners [12]. The consequences of DIVA are significant, with PIVC insertion failure associated with substantial treatment delays [6], increased healthcare costs [13] and significant pain and patient suffering [14,15]. These reasons have most likely been important drivers for the new—Management of PIVCs Clinical Care Standard released by the Australian Commission on Safety and Quality in Health Care, recommending improved monitoring of PIVC outcomes and shared decision making between patients, carers and clinicians [16]. While recommendations on PIVC management are urgently needed to augment care [17], much uncertainty persists in relation to shared decision making in the context of DIVA and how the patient experience can inform future guidelines.

In Australia, current systems fail to measure health outcomes [18–20] and patient and practitioner experience related to DIVA (Schults et al, AHR under review). Thereby the processes which are associated with better outcomes remain unclear and the patient experience largely overlooked. Preliminary work conducted in paediatrics show this is largely due to a lack of supporting infrastructure such as policy and training. For example, inserters have little training and preparation before being asked to insert PIVCs, with a lack of formalised DIVA pathways to support difficult insertions [17]. Further, international guidelines to support ultrasound (US) guided PIVC insertion as the first approach for DIVA patients, [21,22] are lacking [17,23–25]. Despite there being growing evidence to support ultrasound PIVC insertion as the first approach for DIVA patients [26,27], implementation in Australia is negligible [28]. Implementation is challenging as PIVC insertion is not limited to one tightly defined

professional group but rather across professions/specialties. While globally 80% of PIVCs are nurse-inserted, in Australia this is just 20%, with most insertions by junior medical staff [28]. As such our current workforce and systems require purposeful adaptation to implement this capacity. As such health services need to consider adopting implementation strategies based on stakeholder needs relevant to the Behaviour Change Wheel to support sustained behaviour change [29].

With significant demand for PIVC insertion in the context of DIVA, it is likely patients, and the current workforce will become increasingly vulnerable to the negative consequences of PIVC insertion failure without purposeful adaptation of the system to improve capacity. In this context, the present study aimed to understand the experiences of patients and healthcare practitioners (HCPs) with DIVA and PIVC insertion. As most studies to date have focused on the patient experience [15,30], we specifically sought to elucidate the challenges HCPs face, including US use, to inform future studies, interventions, and health care policy development. A secondary aim of the study was to understand what factors may assist in the implementation of new DIVA policy and resources, as mapped to the COM-B.

These objectives informed the following research questions:

1. What are the current and desired approaches to PIVC insertion in patients with DIVA?

2. What are the barriers/enablers for US use?

3. What resources are required to support the sustainable implementation of a clinical pathway for DIVA patients?

4. What are the experiences of Australian patients when undergoing simple and difficult PIVC insertion?

5. What technology and supportive services do Australian patients (patients) want to improve PIVC insertion procedures?

6. When mapped to the Behaviour Change Wheel, how do respondents' experiences with DIVA inform future US implementation strategies?

## Methods

### Design and setting

A descriptive qualitative study was undertaken at healthcare facilities across Australia from 5[th] August 2020 to 15[th] January 2021. We adopted a naturalist philosophy [31] which is concerned with studying something in its natural state rather than applying a specific theoretical perspective. This approach allowed us to develop a more thorough understanding of participants' DIVA experiences, and has been adopted in contemporary, qualitative, health service research [32]. Ethical approval to conduct the study was obtained from Griffith University Human Research Ethics Committee (GU: 2020/157). Participants provided written, informed consent and were able to terminate the interview at any time. Results are reported in line with the Consolidated criteria for reporting qualitative research (COREQ) guidance [33].

**Recruitment and characteristics of participants.** Participants were HCPs responsible for inserting PIVCs across many Australian patient populations and contexts, and patients who had experience with DIVA. We used purposeful sampling [34] and snowballing, to achieve a balanced sample of HCPs with respect to location and discipline. An email seeking study participation was distributed to the research group's professional organisations (e.g., Australian Vascular Access Society; Council of Remote Area Nurses of Australia, Australian College of

Nursing). The lead investigators then approached individual HCPs via email or telephone with a standardised script explaining the details of this voluntary study. To ensure even more perspectives, investigators wrote to a few health services that represented additional geographic diversity and whose workers had not yet been included in the sample. Patients who experienced or whose child experienced difficult PIVC insertion were recruited through invitations to participate via social, radio, online and paper media (e.g., Queensland Country Life (newspaper); Healthcare Awareness Society of Australia [Facebook site], ABC radio interviews, CQ University online news), with additional invitations sent to healthcare patient groups via email. Due to the broad dissemination strategy (used to minimise coverage and sampling error [35] we were not able to calculate a denominator and subsequent response rate.

## Data collection—semi structured interviews

Three interviewers (two research nurses [MF and JK] and one investigator [JS]) received one-on-one training on interview methods and DIVA (derived from existing literature reviews and quality activities [10,17] to carry out in-depth interviews across the vast geographical settings. Experienced moderators (MC and PC) facilitated the training and oversaw the in-depth interview process. The interview guide was informed by prior research conducted by members of the team [15,17,20,28,36] informal discussions with agency leaders, prior studies on DIVA [10,37] and PIVC outcomes [38–40]. Interview questions broadly focused on participants' lived experience with DIVA (supplementary material 1) and included open-ended questioning. Follow-up prompts were designed to lead participants to recount their personal experience with the DIVA and could be adapted based on participant responses during the interview, allowing a more individualised approach [41]. Interviews took place in person, or via telephone due to COVID-19 restrictions on the geographical spread of responding participants. Participant characteristics were noted, as well as the interview setting and conditions, while the interviewer introduced themselves as a clinician researcher working in the field of vascular access. All interviews were audio recorded, with recordings professionally transcribed verbatim for accuracy [42]. Interviews lasted approximately 30 minutes; we did not collect non-verbal data.

Sample size was not defined a priori as we applied the principle of data saturation, where no new themes emerged from interviews [43,44]. Data saturation was determined using field notes taken by the interviewer detailing a summary of the salient points of the interview. The interviewer then summarised the 'perceived' salient points and presented these back to the study participant (on the same day for agreement) to enhance the reliability of study findings. Salient points were then collated contemporaneously to ensure that data saturation was apparent across the multiple interviewers and participants.

## Data analysis

Inductive thematic analysis was used to detail participants' experiences. HCP and patient data were analysed separately. Analysis was undertaken as per Braun and Clarke's six phases of thematic analysis [45]. Initially three researchers (MC, JS, PC) read transcribed interviews and independently generated initial codes. Line-by-line coding was used (facilitating an audit trail) to enhance dependability [46]. Codes were then used to inform concept formation, and themes and sub-themes identified by consensus between researchers. Themes were reviewed in relation to coded extracts and a thematic map generated (led by MC). A selection of extract examples is provided in text to support final themes. Themes were reviewed and defined with continued reference to codes and raw data via discussion with the project team to enhance authenticity [47]. A number of strategies were used to enhance data quality and increase

rigour, including data immersion and triangulation of emerging findings between researchers [48].

Interview themes were mapped to the *Behaviour Change Wheel* [29] by the senior investigator (MC) and cross checked by a second investigator (JS). This implementation theory considers three **sources** of behaviour, ***Capability (psychological and/or physical)***, ***Opportunity (social and/or physical)***, and ***Motivation (autonomic and/or reflexive)–COM-B*** [29]. These sources of behaviour interact to influence and are influenced by other sources of behaviour. Interventions can be designed to address COM-B deficits across all components and as such are multi-faceted. For DIVA PIVC ultrasound implementation, e.g. *physical capability* requires new ultrasound skills, *psychological capability* requires new thought processes to identify DIVA status, *physical opportunity* demands available machines, *social opportunity* requires cultural change that resists repeated landmark attempts, *reflective motivation* may involve internal goalsetting for first attempt success rate, and *automatic motivation* may require belief that ultrasound can achieve first attempt success.

## Results

In total, 78 participants (65 HCPs and 13 patients) across seven Australian states and territories participated Table 1. HCP participants were medical (n = 22; 34%) and nursing (n = 43; 66%) staff working in diverse health care settings from metropolitan facilities (60%) to rural and remote locations (15%). Thematic analysis identified 4 major themes with associated subthemes Table 2. Theme 1 is representative of patients' current experiences with PIVC insertion. Themes 2–4 describe HCPs' experience within this clinical context.

### Theme 1: Harmful patient experiences persist, with patient insights not leveraged to effect change

Patients explained that DIVA insertion disrupted their routine medical treatment and day-today life. Reporting the extent and severity of failed PIVC insertions as '*common*' yet '*horrendous*', with multiple patients recalling feeling '*ignored*' or '*dismissed*' when they identified as having '*DIVA*'. Participants also reported feeling like a '*bad patient*' when they requested an experienced inserter to avoid multiple insertion attempts, and that there was a lack of access to experienced inserters or technology (e.g. US), and were uncertain how to escalate this concern.

> '*Horrendous. Yeah. I've got a chronic disease, which means I go to hospital frequently, both in the public and the private system. Getting someone who is an expert to put a drip in is an absolute debacle*' [RRC2'].

Consequences of repeat insertion attempts were reported to include *scarring, apprehension —fear of next insertion, distress, pain* and *bruising*. For patients in rural and remote settings, extreme coping strategies were discussed with one participant reporting.

> '*When I was younger, I had a lot more issues with cannulations. I would have nurses that would attempt three—even like four or five times to cannulate myself before they would get it. . . since then, I've learned to cannulate myself*' [RRC1].

In addition to worrying about their medical condition, patients also reported feelings of avoidance due to fear of subsequent IV insertions '*I will do anything to avoid going to the Emergency Department*' and when hospital admission was unavoidable participants reported

**Table 1. Summary of participant characteristics (N = 78).**

| Participant characteristic | Healthcare Practitioner N = 65 (%) | Consumer N = 13 (%) |
|---|---|---|
| **Gender** | | |
| Female | 37 (57) | 12 (92) |
| Male | 28 (43) | 1 (8) |
| **State or Territory** | | |
| Queensland | 28 (43) | 10 (77) |
| New South Wales | 21 (32) | 2 (15) |
| Victoria | 6 (9) | 0 |
| South Australia | 6 (9) | 0 |
| Western Australia | 2 (3) | 0 |
| Northern Territory | 1 (2) | 0 |
| Tasmania | 1 (2) | 1 (8) |
| **RRMA Classification** | | |
| Metropolitan | 39 (60) | 6 (46) |
| Regional | 16 (25) | 5 (39) |
| Rural/remote | 10 (15) | 2 (15) |
| **Speciality** | | |
| Nursing and midwifery[a] | 43 (66) | |
| *Nurse practitioner* | *6 (14)* | |
| *Nurse* | *35 (81)* | |
| *Midwife* | *2 (5)* | |
| Medical[$] | 22 (34) | |
| *Resident* | *5 (23)* | |
| *Registrar* | *2 (9)* | |
| *Consultant* | *15 (68)* | |
| **Patient population** | | |
| Adults | 40 (63) | |
| Mixed | 17 (27) | |
| Paediatrics | 2 (3) | |
| Neonates | 4 (6) | |
| Paediatrics & neonates | 1 (2) | |

RRMA: Rural, Remote and Metropolitan Area classification; [$] = total.

**Table 2. Themes and subthemes summarising healthcare practitioners and consumers experiences with DIVA.**

**Theme 1: Harmful patient experiences persist, with consumer insights not leveraged to effect changeSubthemes**
- Feeling invisible
- Risk and anticipation of failed PIVC attempts
- Inflexible processes which don't consider patients' needs

**Theme 2: 'Escalation' is just a word on the front lines.Subthemes**
- Providing day-to-day care for DIVA patients
- Reliance on 'have a go' culture
- Forced to insert PIVCs in an environment lacking resources and support.

**Theme 3: Heightened risk of insertion failure without resources and training.Subthemes**
- Awareness of the benefit of a DIVA pathway
- Education and equipment to support a skilled workforce
- Inserter role and accreditation clarity

**Theme 4: Theme 4: Paving the way forward–'measures need to be in place to prevent failed insertion attempts'. Subthemes**
- System approach including protocols
- DIVA identification processes

'*withdrawing mentally during cannulation as a coping mechanism for pain and discomfort caused by failed insertion attempts*'[RRC1]. Finally, patients discussed the need for a '*national body to support change in local policy and training*'.

## Theme 2: 'Escalation is just a word' on the front lines

HCPs reported that disruptions to medical care from failed insertion attempts were common in hospital settings. Participants discussed varied support and processes for the recognition and assessment of DIVA from (most commonly) no formal process to the use of PIVC insertion policies and finally DIVA decision-making tools. One HCP described the lack of policies meant the '*The intern . . . has a few goes and then comes to find me and gets me to do it (experienced inserter)*' [RI2A]. In facilities that had policies to support PIVC insertion in patients with DIVA, participants suggested this was largely '*ignored*' with recommendations such as 'two attempts' then escalate equating to multiple inserters having two attempts before escalating to an advanced inserter '*it's common for 6 to 8 insertion attempts to be made on neonates before escalating*'. Participants described a reliance on alternative sources for support when DIVA policies were not in place. The consequences of failed insertion attempts concerned HCPs who described feelings of distress and stress when they were unable to cannulate '*A cannula being delayed for several hours might indicate that a patient doesn't get their antibiotics for an infection for many hours and that's more detrimental to the patient*' [R13A].

In metropolitan hospitals, escalation after hours was most frequently to anaesthetics and this often resulted in further delay because of competing priorities for the anaesthetists. Unsurprisingly, escalation in regional and remote settings was discussed as more challenging, with limited access to technology such as US and advanced inserters e.g., anaesthetists. HCPs from remote locations reported '*try(ing) their best and hope(ing) for the best*'. Consequences of failed insertion attempts included escalation to interosseous device insertion or transfer to a larger healthcare facility which may have meant hours in transit and contributed to significant treatment delays. Owing to the lack of formal processes and training, participants discussed the current '*have a go*' culture and the subsequent delay if escalated to a more experienced colleague due to staffing availability. This was further complicated by HCPs who perceived a difficult balance between the need for junior doctors to learn important cannulation skills and limiting insertion attempts by escalation to someone more experienced.

## Theme 3: Heightened risk of insertion failure without resources and training

In general, HCPs believed that '*PIVC insertion in DIVA patients should be attempted by the most experienced . . . clinician first*' or a DIVA team to prevent multiple insertion attempts. HCP participants also described a lack of uncertainty regarding '*whose role it is*' to insert PIVCs and described uncertainty as to when they were '*accredited*' inserters. However, it was noted that a patient deemed difficult for one HCP may not be difficult for another HCP. While some participants reported PIVCs were inserted by both nurses and doctors, many described a perceived reluctance of doctors to escalate PIVC insertion to more experienced nurses. Interestingly some nursing participants worried about whether it fell within their '*scope of practice*'. Staff turnover was also highlighted as an important factor in workforce training considerations with one participant noting '*medical staff frequently rotate or move on*' whilst '*nurses generally stay*'. HCPs in regional, rural and remote settings discussed the ongoing challenges associated with insufficient resources to identify and escalate patients with DIVA, stating '*in the bush just have a go as we have no choice*'. Participants discussed the need to '*provide evidence to decision makers to acquire funding for a DIVA service*' and continued drive to establish '*a vascular access*

*team with sufficient resources'*. However, this change to workforce was believed to be hindered by insufficient resources, and ongoing deficits in education, training and policy and equipment.

HCPs described a lack of formal PIVC insertion training, regulated accreditation and ongoing skill building particularly in terms of technology-assisted capabilities such as US. *'ICUs seem to have their own rules and they don't actually require us to have—they don't actually regulate that you've passed the accreditation process before placing cannulas'* [MI10A]. HCPs also described the *'benefit of US'* for PIVC insertion in patients with DIVA, however due to a lack of policy and resources, US was not used as often as it should, US machines *'are not readily available'*. Overall HCPs reported a need for more formal PIVC insertion courses suggesting *'more training, cannula options and US'* equipment was needed to enable a skilled workforce. Funding and access to resources was also highlighted as a challenge to implementing US for DIVA, with one participant noting *'US ranks low on priority list for small, underfunded healthcare services'*. Interestingly, experienced inserters described difficulty finding a balance *'between (IV insertion) training and patient care'* [MI7A]. The lack of education and support was particularly evident for medical inserters with participants noting *'nurses, they go to a formal education program'* [RI2A] with such a program lacking for medical staff who relied on on-the-job training using a see-one/do-one approach.

### Theme 4: Paving the way forward–*'measures need to be in place to prevent failed insertion attempts'*

HCPs explained that increased advocacy and processes are needed to protect patients with DIVA. A multi-pronged approach was discussed including improved systems and DIVA identification process. HCPs discussed the need for a flagging system such as *'DIVA alert system'* which triggered a clinical pathway including *'improved accessibility to US'*. This would involve having appropriate infrastructure such as US equipment, training, and governance. Some HCPs discussed models of standardised US use for DIVA within discreet departments such as ICU, Emergency Departments or Neonatal ICU. In describing the model of care in place they highlighted some key principles that resulted in success. These included strong leadership that committed over years to training and competence, PIVC policy adherence; early patient assessment and DIVA identification; consideration of the requirement for PIVC and easy access to a well-maintained ultrasound.

HCPs also reported improved levels of support were needed at the policy level to support individual clinicians provide optimal vascular access care across their shift and health settings. Interestingly, shared decision making was highlighted by HCPs as important strategy to highlight in future DIVA policies, with one HCP stating *'staff should have discussions with patients'*. Finally, HCPs discussed *'better preparation'* as important when considering and protecting the patient's long term vessel health and preservation.

### Interview themes mapped to COM-B sources of behaviour

Interview themes and sub-themes were mapped against the COM-B sources of behaviour and intervention functions to inform potential strategies for future DIVA resource implementation [29] (Table 2).

At the policy level, improved DIVA resources are needed with potential expansion of nursing roles, which in turn will increase workforce capability and motivation [10]. US education and training will be key and addresses both physical and social opportunity thus increasing motivation, with training increasing both autonomic and reflexive motivation. Educational

strategies could include point of care resources such as short videos and example scenarios and training with clear and succinct processes for initial and on-going accreditation.

➢ Insert Table 3. Interview themes mapped to COM-B sources of behaviour.

**Table 3. Interview themes mapped to COM-B sources of behaviour.**

| | Themes | Sub-themes | COM-B sources of behaviour (in bold) and Intervention functions (Michie et al, 2011) | Multi-component intervention |
|---|---|---|---|---|
| 1 | The harmful patient experience persists, with consumer insights not leveraged to effect change | Feeling invisible | **Capability (psychological)** Education, training, enablement | **Education of staff** **Education and training of inserters** **Education of patients** |
| | | Risk and anticipation of failed PIVC Attempts | **Opportunity (social)** Restriction, environmental restructuring, enablement | **DIVA identification (e-health record)** |
| | | Inflexible processes which don't consider patients' needs | **Opportunity (physical)** Restriction, environmental restructuring, enablement | |
| 2 | 'Escalation' is just a word on the front lines | Providing day-to-day care for DIVA patients | **Motivation (reflective)** Education, persuasion, incentivisation, coercion | **Education of staff** **Audit of insertion success per ward** |
| | | Reliance on "have a go' culture | **Motivation (reflective)** Education, persuasion, incentivisation, coercion | **Education of staff** **Audit of insertion success per ward** |
| | | Forced to inset PIVCs in an environment lacking resources and support | **Opportunity (physical)** Restriction, environmental restructuring, enablement | **DIVA standard of care and guidelines** |
| 3 | Heightened risk of insertion failure without resources and training | Awareness of the benefit of a DIVA pathway | **Motivation (reflective)** Education, persuasion, incentivisation, coercion | **Education of staff** **Audit of insertion success per ward** |
| | | Education and equipment to support a skilled workforce | **Capability (psychological)** Education, training, enablement **Opportunity (physical)** Restriction, environmental restructuring, enablement | **Education of staff** **Bedside US equipment etc** |
| | | Inserter role and accreditation clarity | **Opportunity (physical)** Restriction, environmental restructuring, enablement | **Initial and ongoing competency** **Accreditation** |
| 4 | Paving the way forward–'measures need to be in place to prevent failed insertion attempts | System approach including protocols | **Capability (psychological) (physical)** Education, training, enablement | **Point-of-care resources and tools– evidence-based DIVA assessment and pathway tool; videos, example scenarios** |
| | | DIVA identification processes | **Capability (psychological)** Education, training, enablement **Capability (physical)** Training, enablement **Opportunity (physical)** Restriction, environmental restructuring, enablement | **Simulation training sessions and clinical skills assessment** **Peer training models** **DIVA standard of care and guidelines** **Evidence-based DIVA assessment and pathway tool** |

DIVA = Difficult intravenous access; PIVC = Peripheral intravenous catheter.

## Discussion

This study achieved its aim to describe the experiences of HCPs and patients regarding DIVA. Our findings suggest that obtaining peripheral vascular access in patients with DIVA is an ongoing clinical issue spanning multiple healthcare settings, with patients often feeling unsupported and invisible. However, HCPs caring for patients with DIVA reported feeling restricted in their abilities to provide care owing to an absence of DIVA policies, resource shortages (US machines) and insufficient trained staff across the 24-hour shift. Across all themes, HCPs expressed anxiety stemming from the consequences of failed PIVC insertions, including pain, trauma, delayed treatment. Concern about obtaining PIVC access in patients with DIVA, exacerbated the workforce's existing vulnerabilities and clinical resource challenges, further impacting clinicians' confidence to provide care. Our findings suggest that the current clinical landscape for DIVA remains largely unchanged since previous international reports [15] with the healthcare system failing to leverage important insights to effect change and improve care.

Another key finding was that, across all healthcare settings the 'have a go' attitude persists. Many HCPs in metropolitan facilities spoke about their facility being a training facility, with limited support to identify or escalate DIVAs and the existence of an unspoken understanding that junior medical staff or ward nurses made the first attempt/s before calling for assistance. This finding may reflect uncertainty with respect to guidelines for DIVA and human resource constraints but may also be reflective of historic medical practices. Further PIVC insertion, or failed insertion is not viewed by the health system as having serious negative outcomes [38,49–51]. Overcoming a traditional and ingrained ethos requires consideration of influencing contextual factors and resource limitations. In rural and remote settings, the current practice of 'just have a go' is particularly endemic due to limited skill mix, the wide scope of health staff (particularly nurses), lack of education and support, and the nature of being self-reliant [52,53]. Escalation pathways for DIVA patients in rural and remote settings would need to educate the most stable and plentiful element of the workforce: nurses. This was a challenge identified in this study due to two reasons: firstly, within the political hierarchy it was not accepted that the nurse should be the escalation point over medical staff; and, secondly, most rural and remote workforces lack stability and have frequent turnover (across disciplines). However, for this type of practice to be changed, support from organisational structure must be evident before clinical change can occur [54]. Further, technology support for distance education in the context of US for DIVA training is complicated by poor internet connectivity and bandwidth and a lack of access to US machines [55].

Some of the trade-offs geographically isolated patients reported to manage their DIVA were akin to those faced by metropolitan patients. However, a number were unique and complicated by geographic isolation and the resource poor environments. These differences led to extreme coping strategies such as self-cannulation or treatment discontinuation. The consequences of being unable to gain PIVC in these settings include facility transfer [56], or to escalation to Intraosseous access [57]. Further, if US equipment was available, it was likely staff were not trained in its use due to education deficits and/or staff turnover. Our findings show patients with DIVA, living in rural and remote areas, feel more vulnerable compared to metropolitan counterparts, and with a perception of limited resources to support PIVC insertion. This experience was mirrored in a systematic review of chronic disease and healthcare access which identified the common elements of *geography* (having to travel long distances to access care), *availability of health professionals* (rural areas lacking staff with specialist skills, or being caught in referral 'games' between metropolitan and rural/remote staff), and *rural culture* (feeling like outsiders in metropolitan environments, wanting to be self-sufficient) [58] as having a negative impact on the patient experience.

A multifaceted approach is needed to develop a solution to the challenges described in this study. Both patient and HCP participants identified a solution would involve several strategies including: DIVA pathways, escalation policies, US-guided PIVC insertion training and accreditation. Overall, the development of DIVA health policy was viewed as essential. Additionally, it is timely to commence discussions of possible versus best practice for those in rural and remote contexts. Due to lack of resources (e.g., stable staffing, wider scope of practice for staff, ability to maintain training requirements with the other mandated training, and physical resources of US machine availability), and processes to manage escalation and consequences of not being able to manage escalations locally need to be considered when developing local policy. The results of this study can be used to inform the development of national DIVA US pathways and associated implementation strategies. However, we have identified several important factors which would impact its successful implementation, such as higher staff turnover in rural and remote settings compared to metropolitan areas [59]. If this is applied to the education approach to manage access to US-guided PIVC then current studies showing a turnover rate of 148% in nursing staff and 80% in Aboriginal health practitioners [60] would negate the ability to service the educational needs of staff in rural and remote settings.

The ultimate impact of a DIVA VA pathway and US uptake depends not only on its effectiveness but also on its reach and uptake in the health system and the extent to which it is implemented with high levels of completeness [61]. A unique finding of our study was the preliminary implementation mapping against The behaviour Change Wheel or COM-B [29] using the interview themes and sub-themes is a systematic framework for identifying multi-facetted strategies to achieve behaviour change that is hopefully sustained overtime. This information can be used in future to develop a logic model to describe the causes and effects (shared relationships including resources, activities, and outputs) of a DIVA pathway incorporating great use of US implementation on desired clinical endpoints. This preliminary mapping provides a systematic process for developing strategies to improve the adoption, implementation and maintenance of a DIVA pathway in healthcare.

Our findings may not be generalizable to all health services due to the qualitative nature of the investigation and potential selection bias. However, we adopted a wide, inclusive sampling technique to capture a broad participant group to enhance transferability of findings [62] across the Australian healthcare setting, however this may not be applicable to international health contexts.

## Conclusion

The findings of this study highlight DIVA patients continue to have a poor healthcare experience in the context of PIVC insertion. Poor standardisation of DIVA assessment, escalation, US use and clinician education across hospitals has contributed to the current rates of dissatisfaction with DIVA services. US-guided insertion of PIVCs is recommended by international guidelines for DIVA patients, and would likely improve the DIVA experience, but uptake in Australia has been sporadic with limited resources and infrastructure to support its ongoing use. Quality and safety improvement opportunities exist to improve the patient with DIVA experience and prevent traumatic insertions. These opportunities primarily situate around the development of new health policy related to DIVA. Further, understanding the barriers and facilitators, particularly from rural and regional health settings, is important for informing future DIVA strategies in these complex populations.

## Acknowledgments

We would like to acknowledge the consumers and clinicians who willingly gave of their time to participate in this study. We would also like to thank Rita Nemeth for help with preparing the manuscript for submission.

## Author Contributions

**Conceptualization:** Jessica A. Schults, Pauline Calleja, Claire M. Rickard, Catriona Booker, Nicole Marsh, Joshua Byrnes, Gerben Keijzers, Marie Cooke.

**Data curation:** Jessica A. Schults, Pauline Calleja, Eugene Slaughter, Claire M. Rickard, Mary Fenn, Jenny Kelly, Marie Cooke.

**Formal analysis:** Jessica A. Schults, Pauline Calleja, Claire M. Rickard, Marie Cooke.

**Funding acquisition:** Jessica A. Schults, Pauline Calleja, Claire M. Rickard, Catriona Booker, Nicole Marsh, Joshua Byrnes, Gerben Keijzers, Marie Cooke.

**Investigation:** Jessica A. Schults, Pauline Calleja, Eugene Slaughter, Claire M. Rickard, Nicole Marsh, Mary Fenn, Jenny Kelly, Peter J. Snelling, Marie Cooke.

**Methodology:** Jessica A. Schults, Pauline Calleja, Eugene Slaughter, Rebecca Paterson, Claire M. Rickard, Catriona Booker, Joshua Byrnes, Gerben Keijzers, Marie Cooke.

**Project administration:** Jessica A. Schults, Pauline Calleja, Eugene Slaughter, Claire M. Rickard, Marie Cooke.

**Writing – original draft:** Jessica A. Schults, Pauline Calleja, Rebecca Paterson, Peter J. Snelling, Marie Cooke.

**Writing – review & editing:** Jessica A. Schults, Pauline Calleja, Eugene Slaughter, Rebecca Paterson, Claire M. Rickard, Catriona Booker, Nicole Marsh, Mary Fenn, Jenny Kelly, Peter J. Snelling, Joshua Byrnes, Gerben Keijzers, Marie Cooke.

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
