## [Decision Letter · Decision Letter 0]

23 Sep 2021

PONE-D-21-22073

Peripheral intravenous catheter insertion and use of ultrasound in patients with difficult intravenous access: Patient and practitioner perspectives to inform future implementation strategies.

PLOS ONE

Dear Dr. Schults,

Thank you for submitting your manuscript to PLOS ONE. After careful consideration, we have decided that your manuscript does not meet our criteria for publication and must therefore be rejected.

I am sorry that we cannot be more positive on this occasion, but hope that you appreciate the reasons for this decision.

Yours sincerely,

Amit Bahl

Academic Editor

PLOS ONE

Reviewers' comments:

Reviewer's Responses to Questions

**Comments to the Author**

1. Is the manuscript technically sound, and do the data support the conclusions?

Reviewer #1: Yes

Reviewer #2: No

2. Has the statistical analysis been performed appropriately and rigorously? 

Reviewer #1: N/A

Reviewer #2: N/A

3. Have the authors made all data underlying the findings in their manuscript fully available?

Reviewer #1: Yes

Reviewer #2: No

4. Is the manuscript presented in an intelligible fashion and written in standard English?

Reviewer #1: Yes

Reviewer #2: No

5. Review Comments to the Author

Reviewer #1: The manuscript is technically sound. Survey was formed and preformed using a systematic approach. The thematic analysis themes that emerged are common in what I see in my practice. The statistical analysis consisted of some percentages of the respondents and that is all that was needed. Complying the themes that emerged does not call for a p-value. The manuscript was written in an intelligent manor and easy to follow.

Reviewer #2: This is a very confused paper with several bias:

1) it reports an experience that - even if true - should be referred exclusively to Australia (and this aspect is not stated, either in the title or in the abstract

2) the selection of healthcare operators and of patients is largely arbitrary; it is likely that different sampling might have yielded different answers

3) it is not known (or not declared) whether all nurses and all patients answering the interview had a clear idea of what 'DIVA' means

4) while a questionnaire might be somehow objective (sometimes) a direct interview leaves much room to non-scientific interpretation of the subject's answers; in other words, it is impossible to state in which degree the answers were 'directed' by the interviewer

5) the overall impression is that this manuscript has few characteristics of a scientific paper; while the topic is obviously important and interesting, the way is addressed and discussed is very 'journalistic' and more apt to be published on a popular magazine than on a scientific journal.

6. PLOS authors have the option to publish the peer review history of their article (what does this mean?). If published, this will include your full peer review and any attached files.

Reviewer #1: No

Reviewer #2: No

- - - - -

---

## [Author Response · Author response to Decision Letter 0]

14 Nov 2021

Thank you for your support of our appeal and the ability to resubmit the manuscript for peer review. 

Response to reviewers’ comments.

Re: PONE-D-21-22073. Peripheral intravenous catheter insertion and use of ultrasound in patients with difficult intravenous access: Patient and practitioner perspectives to inform future implementation strategies.

Thank you for your review of the above manuscript, please find responses to points raised below.

Overall, we would like to point out the lack of feedback and the stark contrast between the positive reviewers of reviewer one and the at times unprofessional review provided by reviewer 2. We request a third review and different academic editor.

Comment Response

Reviewer 1 

The manuscript is technically sound. Survey was formed and preformed using a systematic approach. The thematic analysis themes that emerged are common in what I see in my practice. The statistical analysis consisted of some percentages of the respondents and that is all that was needed. Complying the themes that emerged does not call for a p-value. The manuscript was written in an intelligent manor and easy to follow. Thank you, we would like to highlight the vast difference in feedback from reviewer 1 compared with reviewer 2 yet there was no explanation provided by the editor and we believe the emotive and unprofessional language provided by reviewer two has influenced the decision of the academic editor. 

Reviewer 2 

This is a very confused paper with several bias: We respectfully disagree and have concerns regarding the reviewer’s experience and knowledge of reported methods

It reports an experience that - even if true - should be referred exclusively to Australia (and this aspect is not stated, either in the title or in the abstract We find the comment – even if true- to be disrespectful and not in keeping with the principles of GCP and the purpose of constructive and fair peer review. We are also disappointed this comment was not filtered by the academic editor and therefore request a different editor to handle to appeal. 

However, we have made a change to the title to reflect this concern. 

‘Peripheral intravenous catheter insertion and use of ultrasound in patients with difficult intravenous access: Australian patient and practitioner perspectives to inform future implementation strategies’.

We note however that PLOS One publishes national studies as evidenced through recent publications of the Hungary experience in outpatient care and the Ghana experience of women using reproductive technology. 

the selection of healthcare operators and of patients is largely arbitrary; it is likely that different sampling might have yielded different answers This is a common limitation of opt in sampling. One we acknowledge in the limitations of the paper. However, we used a wide, rigorous sampling strategy one which we discuss in detail in the methods. We did not select participants we offered the interviews to patients and healthcare providers nationally. 

it is not known (or not declared) whether all nurses and all patients answering the interview had a clear idea of what 'DIVA' means The eligibility criteria for interviewees was experience of history with DIVA. This was thoroughly explored and explained during the information and consent process to ensure informed consent and patient eligibility requirements. This comment could be said for any intervention/healthcare experience is an unfair/unsupported statement with the description provided in the methods. 

while a questionnaire might be somehow objective (sometimes) a direct interview leaves much room to non-scientific interpretation of the subject's answers; in other words, it is impossible to state in which degree the answers were 'directed' by the interviewer This is an accepted, expected and routinely reported instrument (interview guide) which is considered best practice in qualitative methods which use interviewing. These methods use an interview guide to standardise questions and reduce bias – a process widely published across disciplines including nursing, medicine, psychology and more. Further the rigour of the study would be reduced without this instrument.

the overall impression is that this manuscript has few characteristics of a scientific paper; while the topic is obviously important and interesting, the way is addressed and discussed is very 'journalistic' and more apt to be published on a popular magazine than on a scientific journal We firmly and wholeheartedly disagree with the reviewer, and in fact are disappointed that this feedback was included by the editor in our collated feedback. 

We believe this viewpoint is biased towards quantitative methods and in direction contrast to PLOS One publishing and review policies.

We would direct you to other manuscripts which use the same methods, reporting framework and tone – which have been previously published in PLOS One and have been well cited and used internationally. 

It is important to note the expert methodologist on paper one is the expert methodologist, senior academic and key writer of this manuscript.

• Not "just" an intravenous line: Consumer perspectives on peripheral intravenous cannulation (PIVC). An international cross-sectional survey of 25 countries

• Experiences of women undergoing assisted reproductive technology in Ghana: A qualitative analysis of their experiences

• Eliciting preferences for outpatient care experiences in Hungary: A discrete choice experiment with a national representative sample

Overall ratings 

1. Is the manuscript technically sound, and do the data support the conclusions?

Reviewer #1: Yes

Reviewer #2: No Again, we would like to highlight the vast difference in feedback from reviewer 1 compared with reviewer 2.

The methods are rigorous conducted using best practice for qualitative methods and reported inline with internationally standardised and accepted reporting frameworks. 

We believe this rating is reflective of reviewer 2 inexperience and lack of understanding of qualitative methods.

As such we request a third review. 

4. Is the manuscript presented in an intelligible fashion and written in standard English?

Reviewer #1: Yes

Reviewer #2: N 

Again, we would like to highlight the vast difference in feedback from reviewer 1 compared with reviewer 2

---

## [Decision Letter · Decision Letter 1]

14 Feb 2022

PONE-D-21-22073R1Peripheral intravenous catheter insertion and use of ultrasound in patients with difficult intravenous access: Patient and practitioner perspectives to inform future implementation strategies.PLOS ONE

Dear Dr. Schults,

Thank you for submitting your manuscript to PLOS ONE. After careful consideration, we feel that it has merit but does not fully meet PLOS ONE’s publication criteria as it currently stands. Therefore, we invite you to submit a revised version of the manuscript that addresses the points raised during the review process.

Please revise.

We look forward to receiving your revised manuscript.

Kind regards,

Academic Editor

PLOS ONE

Journal Requirements:

3. Please amend the title either on the online submission form or in your manuscript so that they are identical. 

Additional Editor Comments (if provided):

Reviewers' comments:

Reviewer's Responses to Questions

**Comments to the Author**

1. If the authors have adequately addressed your comments raised in a previous round of review and you feel that this manuscript is now acceptable for publication, you may indicate that here to bypass the “Comments to the Author” section, enter your conflict of interest statement in the “Confidential to Editor” section, and submit your "Accept" recommendation.

Reviewer #1: All comments have been addressed

Reviewer #3: (No Response)

2. Is the manuscript technically sound, and do the data support the conclusions?

Reviewer #1: Yes

Reviewer #3: No

3. Has the statistical analysis been performed appropriately and rigorously? 

Reviewer #1: N/A

Reviewer #3: I Don't Know

4. Have the authors made all data underlying the findings in their manuscript fully available?

Reviewer #1: Yes

Reviewer #3: No

5. Is the manuscript presented in an intelligible fashion and written in standard English?

Reviewer #1: Yes

Reviewer #3: Yes

6. Review Comments to the Author

Reviewer #1: Response to Review and Response - Reviewer 1

The following comments are my observations according to the PLOS’s Criteria for Publication.

1. The study presents the results of original research

The original content of this research is in the approach. The researchers included both health care providers and patients during the survey. Much of the research conducted around ultrasound use and difficult intravenous access patients are quantitative including number of attempts, failure rates and dwell times. This original research is important because it takes us as readers to the patient experience as well as supporting the difficulties shared by health care workers when approaching venous depleted patient

When studying the experience of either the provider or the patient, the answers are subjective and are not confined to a specific criteria.

2. Results reported have not been published elsewhere

I have not seen research with this design specifically with difficult venous access patients.

3. Experiments, statistics, and other analyses are performed to a high technical standard and are described in sufficient detail.

Using the Behavioral Change Wheel to perform a thematic analysis is a systematic method of interpreting the subjective answers given by health care providers and patients. The Behavioral Change Wheel is described in the text, but a visualization of this would help the reader understand more preciously. The thematic analysis is subjective interpretation process. The researchers gave examples of each theme to help the reader understand the themes. It is not necessary for every comment to be listed. The reader should understand the themes are filtered by the researchers own experience and bias. This is a qualitative study about patient and HCP experiences that requires interpretation not an easily measured outcome like failure.

4. Conclusions are presented in an appropriate fashion and are supported by the data.

It is clear to me from the manuscript that the patients have a poor healthcare experience, there is poor standardization of DIVA assessment that contributes to an ineffective escalation process. Theme one, harmful patient experiences, illustrated how the challenge of difficult intravenous access patients affect the patient experience. The theme “have a go” and “has a few goes and then comes to find me and gets me to do it” is common at the bedside. The clinician is encouraged to stick the patient even when they are not confident about placing the device. The conclusions are presented in an appropriate fashion and are supported by the answers given by the participants.

5. The article is presented in an intelligible fashion and is written in standard English.

No issue.

6. The research meets all applicable standards for the ethics of experimentation and research integrity.

The study was approved by the appropriate board.

7. The article adheres to appropriate reporting guidelines and community standards for data availability.

There are no specific guidelines for collecting subjective survey information. the researchers chose a method to organize the answers and applied it to a framework. This is the best that can be done with this qualitative design. There are no quantitative measurements that can be measured against benchmarking.

I stand by my original recommendation this research should be published.

The survey is by nature subjective, but that does not mean that the study is full of bias. The methods used to evaluate the comments was rigorous. However there has to be room for interpretation.

Second review comments

1. Patient interview size was small. How did you decide on the total n? Is this enough to map the themes?

2. Mapping against COM-B sources of behaviour was not explained thoroughly. What exactly is COM-B and were does it come from? I believe C=Compatibility, O=Opportunity, M=Motivation, but what is B?

Reviewer #3: This is a really confusing manuscript. The objective of this study is weird and no conclusion could be drawn from this study.

7. PLOS authors have the option to publish the peer review history of their article (what does this mean?). If published, this will include your full peer review and any attached files.

Reviewer #1: No

Reviewer #3: No

---

## [Author Response · Author response to Decision Letter 1]

7 Apr 2022

Thank you for the opportunity to revise the manuscript

Please see attached comments in relation to reviewer feedback for the manuscript PONE-D-21-22073R1

Response to Review and Response - Reviewer 1 

The following comments are my observations according to the PLOS’s Criteria for Publication.

1. The study presents the results of original research

The original content of this research is in the approach. The researchers included both health care providers and patients during the survey. Much of the research conducted around ultrasound use and difficult intravenous access patients are quantitative including number of attempts, failure rates and dwell times. This original research is important because it takes us as readers to the patient experience as well as supporting the difficulties shared by health care workers when approaching venous depleted patient

When studying the experience of either the provider or the patient, the answers are subjective and are not confined to a specific criteria. 

Response: Thank you no change made. 

2. Results reported have not been published elsewhere

I have not seen research with this design specifically with difficult venous access patients.

Response: Thank you no change made. 

3. Experiments, statistics, and other analyses are performed to a high technical standard and are described in sufficient detail.

Using the Behavioral Change Wheel to perform a thematic analysis is a systematic method of interpreting the subjective answers given by health care providers and patients. The Behavioral Change Wheel is described in the text, but a visualization of this would help the reader understand more preciously. The thematic analysis is subjective interpretation process. The researchers gave examples of each theme to help the reader understand the themes. It is not necessary for every comment to be listed. The reader should understand the themes are filtered by the researchers own experience and bias. This is a qualitative study about patient and HCP experiences that requires interpretation not an easily measured outcome like failure.

Response: Thank you intended change to come – permission requested to include picture (Behaviour change wheel) within the manuscript awaiting copyright permission Michie et al. 

4. Conclusions are presented in an appropriate fashion and are supported by the data.

It is clear to me from the manuscript that the patients have a poor healthcare experience, there is poor standardization of DIVA assessment that contributes to an ineffective escalation process. Theme one, harmful patient experiences, illustrated how the challenge of difficult intravenous access patients affect the patient experience. The theme “have a go” and “has a few goes and then comes to find me and gets me to do it” is common at the bedside. The clinician is encouraged to stick the patient even when they are not confident about placing the device. The conclusions are presented in an appropriate fashion and are supported by the answers given by the participants.

Response: Thank you no change made. 

5. The article is presented in an intelligible fashion and is written in standard English.

No issue.

Response: Thank you no change made. 

6. The research meets all applicable standards for the ethics of experimentation and research integrity.

The study was approved by the appropriate board.

Response: Thank you no change made. 

7. The article adheres to appropriate reporting guidelines and community standards for data availability.

There are no specific guidelines for collecting subjective survey information. the researchers chose a method to organize the answers and applied it to a framework. This is the best that can be done with this qualitative design. There are no quantitative measurements that can be measured against benchmarking.

I stand by my original recommendation this research should be published.

Matt Gibson

Response: Thank you no change made. 

Response to Review and Response - Reviewer 2 

1. Patient interview size was small. How did you decide on the total n? Is this enough to map the themes?

- refer to reporting guidelines

Response: No change made. Reference to reporting guidelines has been made in text. Please see the following statement contained on page 6, line 144-145

Results are reported in line with the Consolidated criteria for reporting qualitative research (COREQ) guidance.

- appropriate framework for reporting qual work, sample size not defined apriori’

Response: Thank you. This is a very large sample for qualitative work (n=78). Sample size was not defined apriori as recommended for these methods. Instead data saturation is used to determine when an appropriate sample size has been achieved. Therefore the sample size was appropriate to map themes. 

2. Mapping against COM-B sources of behaviour was not explained thoroughly. What exactly is COM-B and were does it come from? I believe C=Compatibility, O=Opportunity, M=Motivation, but what is B?

COM-B referenced – contemporary reference. 

Response: Thank you for your comment. The COM-B reference is a seminal framework and therefore cited in this paper. Since its publication the COM-B has been cited more than 6000 times in research articles (google scholar 02/2022). It is a useful and appropriate framework to situate the research. The COM-B is referenced for readers interested in learning more about the application of the framework. 

This is a really confusing manuscript. The objective of this study is weird and no conclusion could be drawn from this study

Response: Thank you for your comment, however we feel that this is not within keeping with the ethos of fair, equitable, peer review. We have not addressed this comment.

---

## [Decision Letter · Decision Letter 2]

31 May 2022

Peripheral intravenous catheter insertion and use of ultrasound in patients with difficult intravenous access: Australian patient and practitioner perspectives to inform future implementation strategies.

PONE-D-21-22073R2

Dear Dr. Schults,

We’re pleased to inform you that your manuscript has been judged scientifically suitable for publication and will be formally accepted for publication once it meets all outstanding technical requirements.

Kind regards,

Academic Editor

PLOS ONE

Journal Requirements:

3. Please amend the title either on the online submission form or in your manuscript so that they are identical. 

Additional Editor Comments (optional):

Reviewers' comments:

Reviewer's Responses to Questions

**Comments to the Author**

1. If the authors have adequately addressed your comments raised in a previous round of review and you feel that this manuscript is now acceptable for publication, you may indicate that here to bypass the “Comments to the Author” section, enter your conflict of interest statement in the “Confidential to Editor” section, and submit your "Accept" recommendation.

Reviewer #1: All comments have been addressed

Reviewer #4: All comments have been addressed

2. Is the manuscript technically sound, and do the data support the conclusions?

Reviewer #1: (No Response)

Reviewer #4: Yes

3. Has the statistical analysis been performed appropriately and rigorously? 

Reviewer #1: Yes

Reviewer #4: N/A

4. Have the authors made all data underlying the findings in their manuscript fully available?

Reviewer #1: Yes

Reviewer #4: Yes

5. Is the manuscript presented in an intelligible fashion and written in standard English?

Reviewer #1: Yes

Reviewer #4: Yes

6. Review Comments to the Author

Reviewer #1: The interviews is consistent in what I see at the bedside. The response divisions are understandable. Table three is great structure to address the issues with DIVA insertion. The next step is to give specific direction to the recommendation of "education of the staff." Creating a standardized education for these recommendations would be incredible.

Reviewer #4: (No Response)

7. PLOS authors have the option to publish the peer review history of their article (what does this mean?). If published, this will include your full peer review and any attached files.

Reviewer #1: No

Reviewer #4: No

---

## [Editor Report · Acceptance letter]

16 Jun 2022

PONE-D-21-22073R2 

Peripheral intravenous catheter insertion and use of ultrasound in patients with difficult intravenous access: Australian patient and practitioner perspectives to inform future implementation strategies.  

Dear Dr. Schults:

I'm pleased to inform you that your manuscript has been deemed suitable for publication in PLOS ONE. Congratulations! Your manuscript is now with our production department. 

Kind regards, 

on behalf of

Dr. Robert Jeenchen Chen 

Academic Editor

PLOS ONE